# Variety of Serotonin Levels in Pediatric Gastrointestinal Disorders

**DOI:** 10.3390/diagnostics13243675

**Published:** 2023-12-15

**Authors:** Loredana Matiș, Lucia Georgeta Daina, Lavinia Maris, Timea Claudia Ghitea, Daniela Florina Trifan, Ioana Moga, Radu Fodor

**Affiliations:** 1Faculty of Medicine and Pharmacy, Doctoral School of Biomedical Sciences, University of Oradea, 410087 Oradea, Romania; matisloredana@yahoo.com (L.M.); laviniamaris2010@yahoo.com (L.M.); 2Medicine Department, Faculty of Medicine and Pharmacy, University of Oradea, 410068 Oradea, Romania; lucidaina@gmail.com (L.G.D.); trifan.daniela17@yahoo.com (D.F.T.); ioana.moga@yahoo.com (I.M.); 3Pharmacy Department, Faculty of Medicine and Pharmacy, University of Oradea, 410068 Oradea, Romania

**Keywords:** neurotransmitters, serotonin, gastrointestinal disorders, psychobiotics

## Abstract

(1) Serotonin primarily regulates our emotions. A complex process, which includes dysfunctions in gastrointestinal motility and deregulation of the gene responsible for serotonin reuptake (SERT), is implicated in the pathophysiology of irritable bowel syndrome (IBS). This also encompasses changes in intestinal microbiota, the response to stress, the intricate interplay between the brain and the digestive tract, heightened sensitivity to visceral stimuli, and low-grade inflammation. This paper aims to investigate the effectiveness of probiotic therapy in managing gastrointestinal and neuropsychiatric symptoms related to serotonin levels, with a focus on individuals with serotonin deficiency and those with normal serotonin levels experiencing gastrointestinal disorders. (2) The study involved 135 pediatric patients aged 5–18 years with gastrointestinal disturbances, including constipation, diarrhea, and other symptoms, such as nausea, flatulence, feeling full, or gastrointestinal pain. (3) Serotonin testing was performed, and administering probiotics appeared to be effective in addressing serotonin deficiency and other gastrointestinal disorders. (4) Serotonin’s pivotal role in regulating neurotransmitter secretion and its impact on neuropsychiatric health, coupled with gender differences and age-related declines, underscore the complexity of their influence on gastrointestinal and neuropsychiatric conditions.

## 1. Introduction

The interaction between the central nervous system and the intestinal environment relies on neural, hormonal, immune, and metabolic responses. In recent decades, significant attention has been devoted to investigating the relationship between the digestive system and the brain. Termed the “gut-brain axis”, this distinctive connection between the gastrointestinal tract and the central nervous system involves a two-way exchange between the two [1].

Serotonin plays a central role in regulating our emotions and works in collaboration with adrenaline and dopamine to elevate mood and control motivation [1]. It also possesses calming effects, enhances sleep quality, and acts as an antidepressant. Furthermore, serotonin is involved in regulating satiety and pain sensitivity and influencing essential intestinal functions and nutrient absorption [2].

Serotonin levels are correlated with diarrhea, with its imbalance contributing to its exacerbation [3]. A significant link has also been established with other gastrointestinal problems [4]. Additionally, a connection has been identified between neuropsychiatric disorders and serotonin levels in individuals with severe chronic gastrointestinal diseases, suggesting a significant relationship [5]. Building upon these results, our hypothesis is that early identification of serotonin imbalance can be instrumental in preventing the development of chronic gastrointestinal diseases.

Irritable bowel syndrome (IBS) involves a complex process influenced by factors such as issues with gut movement, changes in the serotonin reuptake gene (SERT), gut microbiota alterations, stress responses, interaction between the brain and the digestive tract [6], heightened sensitivity to visceral stimuli, and the presence of low-grade inflammation [7]. Gut serotonin, also known as 5HT, governs gut motility, secretions, visceral sensitivity, and inflammatory processes [8].

Genetic abnormalities causing disruptions in enzyme functions are common triggers for neurometabolic diseases, impacting the normal development and function of the nervous system. Although individual disorders are rare, the collective occurrence of neurometabolic diseases is relatively common, posing lifelong challenges that significantly impact society. Neuropsychiatric symptoms resembling ADHD can manifest in numerous neurometabolic diseases, even when the primary biochemical malfunction occurs in cells and tissues outside the nervous system [9]. The link between neuropsychiatric and gastrointestinal disorders is highlighted by the involvement of the microbiota in coexisting intestinal disorders [10].

Depending on the presenting symptoms, various pharmacological options are available, such as laxatives, antispasmodics, serotonergic agents, antidepressants, loperamide, rifaximin, and cholestyramine, alongside the use of herbal therapies. The intricate pathophysiology of IBS highlights the interconnected disturbances within this condition.

Research in the field of the gut microbiome and mental health is continually evolving, and greater emphasis is required to attain a more comprehensive and detailed understanding. Future studies should tackle these challenges by utilizing larger sample sizes, conducting longitudinal investigations, and employing rigorous experimental methodologies. These strategies will enable a more precise grasp of the intricate association between the gut microbiome and mental health and have the potential to pave the way for the development of novel therapeutic interventions in this realm [11].

Here, we investigate the efficacy of probiotic therapy in managing gastrointestinal and neuropsychiatric symptoms associated with serotonin levels, with a focus on individuals with serotonin deficiency and those with normal serotonin levels experiencing gastrointestinal disorders. We focus on its impact on constipation and diarrhea specifically. This paper’s purpose is to establish a connection between gastrointestinal symptoms and potential serotonin imbalances to preemptively address disease progression even before the onset of IBS. The future perspective involves systematically examining the primary effects of neurotransmitters on gastrointestinal and neuropsychic symptoms. This approach aims to develop a more accurate understanding of both gastrointestinal issues and neuropsychic disorders by unraveling the connections between them, with a focus on neurotransmitters.

## 2. Materials and Methods

Between 2020 and 2022, a prospective study was conducted at a private nutrition practice in accordance with the guidelines outlined in the World Medical Association’s Declaration of Helsinki. The patients visited the private medical nutrition clinic “Echo Laboratoare” (Oradea, Romania) with their legal representative, reporting gastrointestinal issues (diarrhea, constipation, and various gastrointestinal disturbances, such as nausea, a sensation of fullness, flatulence, and belching). A clinical assessment was conducted.

Inclusion criteria: The study included patients aged between 5 and 18 years who presented with gastrointestinal problems, including diarrhea, constipation, and other gastrointestinal disorders such as nausea, feelings of fullness, flatulence, and belching.

Exclusion criteria: Patients over 18 years of age and those who declined to participate in the study were excluded. Additionally, patients with other chronic diseases that could potentially impact the study results were excluded. Furthermore, patients with chronic gastrointestinal conditions, such as gastritis, reflux disease, irritable bowel syndrome, or inflammatory bowel diseases, were also excluded due to the potential for interference with neurotransmitter levels resulting from allopathic treatment. Patients with other chronic diseases requiring allopathic treatment were also excluded.

### 2.1. Participant Selection

The study was centered on individuals aged between 5 and 18, all of whom had been diagnosed with metabolic syndrome. Out of 1145 individuals screened, 135 were selected to participate and underwent monthly consultations following the established study protocols. Exclusions from the study were individuals over the age of 18, those who declined to participate, and those with chronic conditions that could potentially influence the study’s outcomes. The required sample size was determined using the appropriate formula for this type of research, resulting in a specified minimum of 85 cases to achieve a 95% confidence level. The research subjects were divided into 3 groups:Control group (I): 37 patients (27.4%).Group with psychoanxiety disorders (II): 65 patients (48.1%).Group with psychiatric disorders (III): 33 people (24.4%).

The distribution of demographic data by group shows an approximately equal distribution with no significant differences in terms of gender (*p* = 0.177), age (*p* = 0.063), or background (*p* = 0.502).

All patients presented with gastrointestinal disturbances, including constipation and diarrhea, as well as additional symptoms such as nausea, flatulence, satiety, or gastrointestinal pain. The patients adhered to a healthy diet (reducing excess intake of sugar, salt, and fried foods) without any clinical dietary interventions. They were prescribed personalized probiotic treatments tailored to address their specific gastrointestinal issues. These recommended probiotics contained various combinations and proportions of *bifidobacteria*, *lactobacilli*, and *saccharomyces* ssp., with formulations excluding gluten or milk, for 3 months.

### 2.2. Clinical Assessment

The clinical evaluation took place in the medical office, where symptoms including constipation, diarrhea, mood changes, hyperactivity, aggressiveness, sleep disturbances, lack of concentration, headaches, fatigue, depression, anxiety, and gastrointestinal issues unrelated to constipation or diarrhea were monitored. A comprehensive patient history was conducted, which included identifying personal medical history, drug use, tobacco use, alcohol consumption, and the use of other restricted substances.

### 2.3. Paraclinical Assessment

To confirm the diagnoses, paraclinical assessments were performed. These assessments included the analysis of neurotransmitter imbalances, particularly of serotonin. Normal serotonin levels range from 100–225 µg/g creatinine [12,13]. These analyses were carried out in an analytical laboratory using enzymatic, colorimetric, and spectrophotometric techniques, along with immuno-enzymatic tests. Assessments were conducted at the beginning and end of the study and underwent peer review. Specific tests utilizing urine and saliva samples were employed to measure the presence of stress hormones in the body (CTL and Ortholabor GmbH, 26160 Bad Zwischenahn, Germany). Serotonin levels are higher up to the age of 5 compared with adults, but after the age of 5, the serotonin level falls within the mentioned range [14].

### 2.4. Statistical Analysis

The study involved an analysis of variations in biomarkers throughout the research period. Numerical and graphical summaries of individual case profiles were generated, considering changes from the baseline. Biomarker distributions did not exhibit deviations from normality. Changes over time were modeled using a random-effects linear mixed model with an unstructured correlation for repeated measures. Time of testing was initially introduced as a categorical variable to compare mean changes and subsequently as a continuous variable to assess temporal trends in biomarkers. Relationships between biomarkers were explored through Spearman’s correlations. Statistical analyses were conducted using SPSS software (version 20) with a significance threshold of *p* < 0.05, and model fit was evaluated for each biomarker at each time point by examining residuals.

### 2.5. Ethics Committee Approval

The study adhered to the guidelines established in the Declaration of Helsinki by the World Medical Association and obtained approval from the institution (Approval No. CEFMF/3, dated 30 October 2023). Each patient was scheduled for at least one monthly visit to the doctor’s office, and assessments were conducted at both the study’s commencement and its conclusion.

## 3. Results

### 3.1. Demographic Description of Patients Based on Serotonin Level Assessment Is as Follows

Serotonin levels below 100 µg/g creatinine were found in 10 subjects (37.0%) in group I and 17 subjects (63.0%) in group II. No patients in group III had serotonin levels below 100 µg/g creatinine. Normal serotonin levels (ranging from 100–225 µg/g creatinine) [12] were observed only in groups I and III, and high serotonin levels, exceeding 225 µg/g creatinine, were predominantly observed in group II (48 individuals, 88.9%), with only 6 individuals (11.1%) in group III exhibiting high levels.

It was noted that low serotonin levels (<100 µg/g creatinine) were present exclusively in male patients, whereas elevated serotonin levels were only found in female patients. These outcomes may be influenced by the relatively small number of patients and could also be correlated with the prevalence of certain diseases based on gender, as detailed in Figure 1.

The average age for the group with serotonin levels below 100 µg/g creatinine was 17.93 ± 0.38 years. In the group with serotonin levels between 100 and 225 µg/g creatinine, the subjects had a mean age of 12.56 ± 4.53 years, and in the group with serotonin levels above 225 µg/g creatinine, the average age was 9.93 ± 2.72 years. A noticeable reduction in serotonin levels is evident with advancing age. This decline may be linked to escalating stress levels in children, which tend to increase as they grow older.

### 3.2. Incidence of Gastrointestinal Symptoms According to Serotonin Levels

After evaluating gastrointestinal symptoms at the end of the research period, it became evident that the incidence of constipation was significantly higher in individuals with serotonin imbalances, encompassing both those with low levels below 100 µg/g creatinine and elevated levels exceeding 225 µg/g creatinine serotonin. Subsequent Bonferroni Post Hoc analysis failed to reveal any noteworthy differences in constipation among the three serotonin groups.

Concerning diarrhea, a statistically significant difference emerged between the groups with serotonin levels <100 µg/g creatinine and 100–225 µg/g creatinine, as well as between the group with serotonin levels <100 µg/g creatinine and the group with levels >225 µg/g creatinine (*p* < 0.05). However, no significant difference was noted between the group with serotonin levels of 100–225 µg/g creatinine and the group with levels >225 µg/g creatinine (*p* > 0.05).

Gastrointestinal problems at the conclusion of the study period displayed significant variations between the groups with serotonin levels <100 µg/g creatinine and >225 µg/g creatinine (*p* < 0.05), as well as between the groups with levels 100–225 µg/g creatinine and >225 µg/g creatinine (*p* < 0.05). No significant differences were observed between the groups with serotonin levels <100 µg/g creatinine and 100–225 µg/g creatinine. Detailed results are provided in Table 1.

In Figure 2, you can observe the progression of symptoms based on serotonin levels. The most notable differences in parameters related to gastrointestinal symptoms were identified in individuals with serotonin deficiency. This may be attributed to the specific action of these probiotics, known as psychobiotics, aimed at rebalancing serotonin. Interestingly, diarrhea and constipation did not show improvement following probiotic therapy. Instead, other gastrointestinal issues, such as nausea, feelings of fullness, flatulence, or belching, exhibited improvement after the administration of psychobiotics. In the case of excess serotonin, specific probiotic therapy was found to be effective in addressing constipation and other gastrointestinal problems.

Following the application of the Pearson statistical test to investigate the correlations between serotonin levels and the advancement of gastrointestinal symptoms, a substantial negative association emerged between constipation and serotonin levels. This inverse relationship is apparent from the negative value of the Pearson coefficient and the “*p*” value, signifying statistical significance. In simpler terms, as serotonin levels increase, the occurrence of constipation decreases, as depicted in Figure 3.

Furthermore, upon employing the Pearson statistical test to examine the relationships between serotonin levels and the development of gastrointestinal symptoms, a notable positive correlation was established between diarrhea, gastrointestinal problems, and serotonin levels. This direct association is apparent from the positive value of the Pearson coefficient and the statistical significance represented by *p* < 0.05. Essentially, as serotonin levels elevate, the occurrence of diarrhea and gastrointestinal problems also increase. The Pearson correlation of serotonin levels and gastrointestinal symptoms is outlined in Table 2.

### 3.3. Incidence of Neuropsychiatric Symptoms Based on Serotonin Levels

When examining neuropsychiatric symptoms at the beginning of the research period, it was observed that the incidence of headaches was significantly higher in individuals with low serotonin levels below 100 µg/g creatinine. Post Bonferroni Post Hoc analysis revealed no significant differences in constipation between the three serotonin groups. However, in terms of headaches, a statistically significant difference was found between the <100 µg/g creatinine and 100–225 µg/g creatinine groups, as well as between the <100 µg/g creatinine group and the >225 µg/g creatinine group (*p* < 0.05). There was also a significant difference between the 100–225 µg/g creatinine groups and the >225 µg/g creatinine group (*p* < 0.05).

Concerning fatigue, hyperactivity, and aggression at the study’s outset, no significant differences were observed between the <100 µg/g creatinine and 100–225 µg/g creatinine groups, as well as between the <100 µg/g creatinine group and the >225 µg/g creatinine group (*p* > 0.05). However, notable differences were noted between the 100–225 µg/g creatinine groups and the >225 µg/g creatinine group (*p* < 0.05).

Mood changes at the study’s conclusion exhibited statistically significant differences among each study group, detailed in Table 3. Regarding sleep disturbances, no significant differences were observed between the <100 µg/g creatinine and >225 µg/g creatinine groups (*p* > 0.05), as well as between the 100–225 µg/g creatinine group and the >225 µg/g creatinine group (*p* > 0.05). Nonetheless, a significant difference was found between the <100 µg/g creatinine and 100–225 µg/g creatinine groups (*p* < 0.05). In terms of concentration deficits, a statistically significant difference was observed between the <100 µg/g creatinine and >225 µg/g creatinine groups (*p* < 0.05). However, no significant differences were found between the <100 µg/g creatinine and 100–225 µg/g creatinine groups, as detailed in Table 3.

After employing the Pearson statistical test to scrutinize the correlations between serotonin levels and the progression of neuropsychiatric symptoms, a robust positive and statistically significant association was identified between headache, fatigue, mood swings, aggression, as well as sleep disorders, and the level of serotonin. This direct relationship is evident from the positive value of the Pearson coefficient and the statistical significance represented by *p* < 0.05. Simply put, as serotonin levels increase, the incidence of headaches, fatigue, mood swings, aggression, and sleep disorders also increases. The Pearson correlation between serotonin levels and differences in neuropsychiatric symptomatology is delineated in Table 4.

## 4. Discussion

Cognitive dysfunctions correlated with neurotransmitter variations, especially serotonin, were discussed by Kalai [15]. He observed the role of the microbiota in regulating cognitive functions, marking an important step in understanding the connection between the gut and the brain [15]. In our study, we explored not only the modulation of the microbiota through probiotic therapy but also its modulation through healthy dietary interventions. It is assumed that food, particularly ultra-processed food, plays an important role in modulating the microbiome, with significant expectations related to achieving balance. According to these expectations, we observed a significant improvement in patients’ attention at the end of the research period.

The increasing prevalence of eating disorders has become a significant global health concern, impacting not just adults but also children and adolescents. Moreover, the accumulation of overall adiposity is closely tied to neurotransmitter imbalances. Furthermore, the accumulation of overall adiposity [16], subcutaneous trunk fat, visceral fat, and intramuscular fat during adolescence has been linked to a heightened risk of atherosclerosis in adulthood, both positively and independently [17].

The various functions of serotonin in the intestinal mucosa have been discussed, with its imbalance being correlated with the inflammatory process [18]. An important role of serotonin in preventing the development of metabolic syndrome has been observed [19]. Serotonergic imbalance has been correlated with the onset of metabolic syndrome, emphasizing its significant role in prevention [19]. Furthermore, serotonin imbalance can be associated with inappropriate eating behavior, which, in turn, is correlated with the development of metabolic syndrome [20]. At the same time, the presence of metabolic syndrome (MS) contributes to inflammation, increased susceptibility to infections [21,22], lower vitamin D levels [23], and a compromised immune response [24]. It is imperative to manage MS through the available approaches, including dietary therapy [25], probiotic therapy [26,27], and physical activity [28], as they represent essential means for addressing this complex health condition. In our study, we focused on pediatric patients while also addressing the complications associated with metabolic syndrome in this specific population.

Gastrointestinal disorders are intricately associated with psychoanxiety disorders. This correlation can be ascribed to the existence of discomfort, pain, and the challenge of controlling these symptoms. Numerous studies have underscored the pivotal role of the intestinal microbiome in addressing psychoanxiety disorders [29,30,31,32,33,34,35,36]. Constipation is a prevalent issue in children with neurological disorders, necessitating a comprehensive approach to address the underlying mechanisms [37,38]. In our current study, the combination of probiotics and dietary therapy proved effective in individuals with psychoanxiety disorders.

The implication is that neurotransmitters play a significant role in diarrhea [39], both in inflammatory bowel diseases [40] and neuropsychiatric imbalances [41]. A balanced diet combined with probiotics can enhance the quality of life, regulate body weight, and promote digestive and neuropsychiatric health [42,43,44,45,46]. We observed that the control group responded best to dietary and probiotic therapy in terms of gastrointestinal disorders. These results can serve as valuable guidelines for effectively managing minor gastrointestinal problems using these methods. In our study, the group with neuropsychiatric disorders showed less improvement compared to the anxiety disorder group, consistent with findings from specialized studies.

Corticotropin-releasing factor triggers the release of adrenocorticotropic hormone (ACTH), which, in turn, activates the adrenal cortex, leading to the synthesis and secretion of glucocorticoids like cortisol. The sudden increase in cortisol and other glucocorticoids initiates mineralocorticoid receptors in the pituitary and hypothalamus, reducing the release of CRF and establishing a negative feedback system. This results in decreased ACTH secretion from the anterior pituitary and, consequently, lower glucocorticoid release [47]. In our current study, elevated morning cortisol levels exhibited positive correlations with headaches and aggressive behavior.

Probiotics with a specific impact on the central nervous system, known as “psychobiotics”, have been the subject of significant discussion in recent years [48]. Evidence of benefits in alleviating symptoms of depression and chronic fatigue syndrome are emerging. These benefits may be associated with the anti-inflammatory actions of certain psychobiotics and their capacity to reduce the activity of the hypothalamus–pituitary–adrenal axis [27,49,50,51], with increasing clarity regarding their effectiveness. In our study, we monitored the progression of neuropsychiatric symptoms during probiotic treatment based on serotonin levels.

The study has several limitations, including the relatively small sample size and the presence of gastrointestinal disorders among the participants. Another limitation to consider is the age factor, which could yield different results from those in the adult population. Additionally, the inclusion of patients diagnosed with mental illnesses and undergoing treatment with SSRIs or SNRIs may potentially obscure the effects of diet therapy and psychobiological therapy.

One notable strength of this study lies in its exploration of the correlations between neuropsychological issues, gastrointestinal problems, and neurotransmitter levels. This understanding can pave the way for more effective regulation, particularly in children, to ensure healthy and harmonious development.

## 5. Conclusions

Serotonin plays a crucial role in regulating neurotransmitter secretion and is linked to catecholamines. Serotonin deficiency affects catecholamines and their ratio, impacting neuropsychiatric health significantly.

Gender differences in serotonin levels, with higher levels in women, align with genetic associations in neuropsychiatric disorders. Considering gender-specific factors is essential in understanding these conditions.

The decrease in serotonin with age may be related to changing dietary habits, especially increased consumption of ultra-processed foods, contributing to declining serotonin levels over time.

High serotonin levels correlate with reduced constipation but increased diarrhea and other gastrointestinal issues, influenced by the nervous system’s control of intestinal functions. Further research is needed for a complete understanding.

## Figures and Tables

**Figure 1 diagnostics-13-03675-f001:**
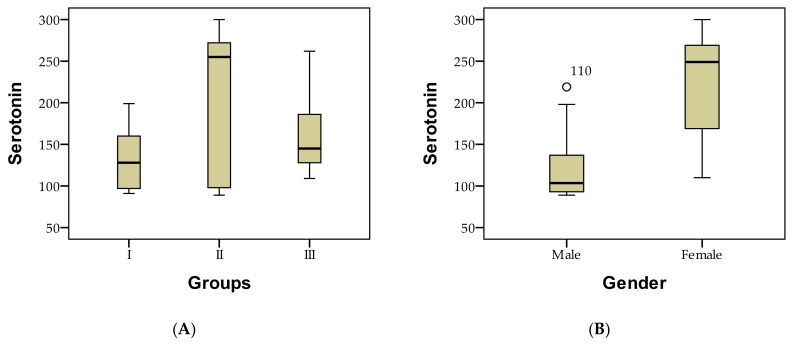
Demographic distribution of patients based on serotonin-level assessment in case of study groups (**A**), gender (**B**), age (**C**), environment of origin (**D**), where ‘*’ represents only one exceptional case, and ‘°’ represents more isolated cases.

**Figure 2 diagnostics-13-03675-f002:**
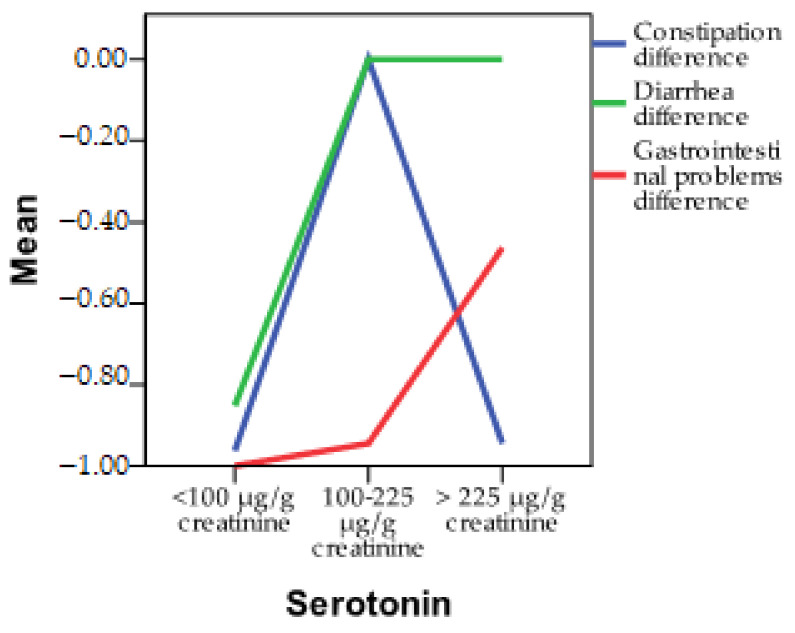
Differences in the progression of gastrointestinal symptoms based on serotonin levels.

**Figure 3 diagnostics-13-03675-f003:**
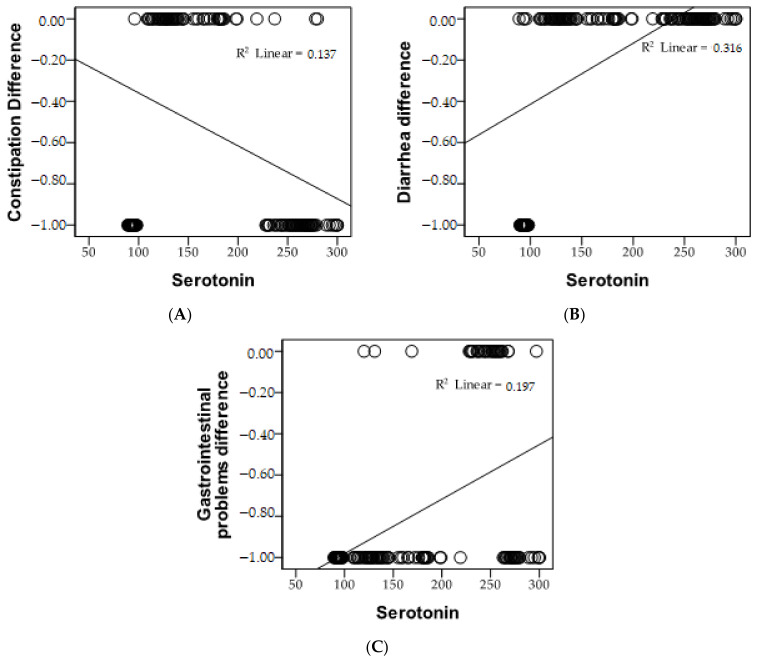
Graphical presentation of the correlations between serotonin levels and the progression of gastrointestinal symptoms, as constipation (**A**), diarrhea (**B**), and gastrointestinal problems (**C**).

**Table 1 diagnostics-13-03675-t001:** Assessment of gastrointestinal symptoms in the three research groups based on serotonin levels.

Parameters		Serotonin Levels						*p*
		<100 µg/g Creatinine		100–225 µg/g Creatinine		>225 µg/g Creatinine		
		*n*	%	*n*	%	*n*	%	
Initial								
Constipation	No	0	0.0	54	100.0	0	0.0	^a^
	Yes	27	100.0	0	0.0	54	100.0	
Diarrhea	No	0	0.0	54	100.0	54	100.0	^a^
	Yes	27	100.0	0	0.0	0	0.0	
Gastrointestinal problems	No	0	0.0	0	0.0	0	0.0	^a^
	Yes	27	100.0	54	100.0	54	100.0	
Final								
Constipation	No	26	96.3	54	100.0	51	94.4	0.231
	Yes	1	3.7	0	0.0	3	5.6	
Diarrhea	No	23	85.2	54	100.0	54	100.0	0.001 **
	Yes	4	14.8	0	0.0	0	0.0	
Gastrointestinal problems	No	27	100.0	51	94.4	25	46.3	0.001 **
	Yes	0	0.0	3	5.6	29	53.7	

*n* = number of patients, *p* = statistical significance, ** = Correlation is significant at the 0.01 level, ^a^ = statistical test cannot be performed.

**Table 2 diagnostics-13-03675-t002:** Pearson correlation between serotonin levels and gastrointestinal symptom variations.

Pearson Correlation		Serotonin
Constipation	r	−0.370 **
	*p*	0.000
Diarrhea	r	0.562 **
	*p*	0.000
Gastrointestinal problems	r	0.444 **
	*p*	0.000
	*n*	135

*n* = number of patients, *p* = statistical significance, r = Pearson’s coefficient, ** = Correlation is significant at the 0.01 level.

**Table 3 diagnostics-13-03675-t003:** Neuropsychiatric symptomatology at the start and end of the study in the three research groups based on serotonin levels.

Parameters		Serotonin						*p*
		<100 µg/g Creatinine		100–225 µg/g Creatinine		>225 µg/g Creatinine		
		*n*	%	*n*	%	*n*	%	
Initial								
Headache	No	0	0.0	27	50.0	27	50.0	0.001 **
	Yes	27	100.0	27	50.0	27	50.0	
Fatigue	No	0	0.0	27	50.0	54	100.0	^a^
	Yes	27	100.0	27	50.0	0	0.0	
Mood swings	No	0	0.0	0	0.0	0	0.0	0.001 **
	Yes	27	100.0	54	100.0	54	100.0	
Hyperactivity	No	27	100.0	0	0.0	27	50.0	0.001 **
	Yes	0	0.0	54	100.0	27	50.0	
Aggression	No	0	0.0	0	0.0	27	50.0	0.001
	Yes	27	100.0	54	100.0	27	50.0	
Sleep disturbances	No	0	0.0	0	0.0	27	50.0	0.001 **
	Yes	27	100.0	54	100.0	27	50.0	
Lack of concentration	No	0	0.0	27	50.0	0	0.0	0.001 **
	Yes	27	100.0	27	50.0	54	100.0	
Final								
Headache	No	27	100.0	54	100.0	49	90.7	0.020 *
	Yes	0	0.0	0	0.0	5	9.3	
Fatigue	No	25	92.6	54	100.0	54	100.0	0.017 *
	Yes	2	7.4	0	0.0	0	0.0	
Mood swings	No	23	85.2	54	100.0	0	0.0	0.001 **
	Yes	4	14.8	0	0.0	54	100.0	
Hyperactivity	No	27	100.0	27	50.0	47	87.0	0.001 **
	Yes	0	0.0	27	50.0	7	13.0	
Aggression	No	27	100.0	27	50.0	48	88.9	0.001 **
	Yes	0	0.0	27	50.0	6	11.1	
Sleep disturbances	No	27	100.0	53	98.1	48	88.9	0.037 *
	Yes	0	0.0	1	1.9	6	11.1	
Lack of concentration	No	27	100.0	54	100.0	27	50.0	0.001 **
	Yes	0	0.0	0	0.0	27	50.0	

*n* = number of patients, *p* = statistical significance, * = Correlation is significant at the 0.05 level, ** = Correlation is significant at the 0.01 level, ^a^ = the statistical test cannot be performed.

**Table 4 diagnostics-13-03675-t004:** Pearson correlation regarding serotonin levels and differences in neuropsychiatric symptomatology.

Pearson Correlation		Serotonin
Headache	r	0.417 **
	*p*	0.000
Fatigue	r	0.682 **
	*p*	0.000
Mood swings	r	0.831 **
	*p*	0.000
Hyperactivity	r	−0.068
	*p*	0.432
Aggression	r	0.426 **
	*p*	0.000
Sleep disturbances	r	0.709 **
	*p*	0.000
Lack of concentration	r	0.162
	*p*	0.060
	*n*	135

*n* = number of patients, *p* = statistical significance, r = Pearson’s coefficient, ** = Correlation is significant at the 0.01 level.

## Data Availability

All the data processed in this article are part of the research for a doctoral thesis, being archived in the aesthetic medical office, where the interventions were performed.

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
