# Peer review of "Variety of Serotonin Levels in Pediatric Gastrointestinal Disorders"

_diagnostics, 2023, doi:10.3390/diagnostics13243675_

Round 1

Reviewer 1 Report

Comments and Suggestions for Authors

The study is actual because the gut-brain interaction is the main point of pathogenesis of the
development of various Childhood Functional Gastrointestinal Disorders (FGIDs), the most
common children GIT-problem. One of the notable point of this study is the research of
correlation of neuropsychological problems, gastrointestinal disorders (GIT-disorders) and
neurotransmitter levels. An understanding of this effective regulation can play the important role in a healthy and harmonious development of children.
The article contains a number of features that requires clarifications and modifications.
The structure of the abstract does not contain the special parts typical for practical work:
materials and methods, practical part and conclusion.
The design of the study in the article is not clear enough: the principle of dividing the subjects
into 3 groups is unclear.
The inclusion/exclusion criteria are questionable, since the inclusion criteria apparently included various childhood FGIDs (its recommended to show the compliance with Rome IV Diagnostic Criteria in the Materials and Methods part). Сontrariwise the IBS, the most common children FGIDs (over 4 years old), is an exclusion criterion.
Materials and Methods part indicates the determination of adrenaline, norepinephrine, dopamine levels in the subjects, but results are not presented. The validity of serotonin/creatinine score values references of in childhood (in different age groups) should be presented. Given the large age differences, it is necessary to provide data for different age periods. Moreover, serotonin levels were significantly higher in the first 12-13 years of life than in adolescents.
Metabolic syndrome in children is not clarified in the data: what is the patients BMI (allowed to
be classified as metabolic syndrome).The role of metabolic syndrome in the study is unclear: are there some correlation of serotonin GIT level and clinical gastrointestinal and neuropsychiatric manifestations (data also should be presented).
Researchers stated in the study the exploration the modulation of the microbiota through
probiotic therapy but also its modulation through dietary interventions. So no any data about
microbiota and diet in the study.
The polyprobiotic complex, diet used for intervention, are not clearly presented by researchers:
data about strains of probiotics used with a proven psychobiotic effect (included in the
composition), duration of therapy are not presented in the study.
The results of the study part presents data on the level of serotonin in the groups of
gastrointestinal manifestations and neuropsychiatric manifestations, but the data of adrenaline-norepinephrine-dopamine.
The discussion of the results of the study is not connected with the practical part ; there are no
clinical parallels of the data obtained on serotonin levels in the intestine and clinical
manifestations with analogues in the studies, no data about elevated morning cortisol levels
exhibited positive correlations with headaches and aggressive behavior in the practical part.
It is recommended to modificate and finalize the article with subsequent publication.

Author Response

Reviewer 1

Firstly, we, the authors of the present manuscript wish to thank you for thoughtful commentary you have provided to improve the quality of the paper. We are very grateful for the time and effort you have devoted to this task. We have extensively revised my manuscript according to the recommendations. All changes in the text and the new figures that we have redesigned are highlighted. Please, see the point-by-point answers to your comments below. All correction was highlighted in the manuscript.

The study is actual because the gut-brain interaction is the main point of pathogenesis of the
development of various Childhood Functional Gastrointestinal Disorders (FGIDs), the most
common children GIT-problem. One of the notable point of this study is the research of
correlation of neuropsychological problems, gastrointestinal disorders (GIT-disorders) and
neurotransmitter levels. An understanding of this effective regulation can play the important role in a healthy and harmonious development of children. The article contains a number of features that requires clarifications and modifications.

  1. The structure of the abstract does not contain the special parts typical for practical work:
    materials and methods, practical part and conclusion.

Answer 1. Thank you very much for observation. We, the authors, I have delimited the sections more visibly.

  1. The design of the study in the article is not clear enough: the principle of dividing the subjects
    into 3 groups is unclear.

Answer 2: Thank you for observation. I completed with detailed description. (lines 105-111)

  1. The inclusion/exclusion criteria are questionable, since the inclusion criteria apparently included various childhood FGIDs (its recommended to show the compliance with Rome IV Diagnostic Criteria in the Materials and Methods part). Ð¡ontrariwise the IBS, the most common children FGIDs (over 4 years old), is an exclusion criterion.

Answer 3. Thank you for amendment. We are well aware of the subtle distinction between disorders and diseases. Therefore, we made a clarification and underscored that we specifically tracked individuals with disorders, excluding those with diseases. This approach enables us to assess neurotransmitter levels from the initial indications of gastrointestinal disorders. I emphasized this distinction more emphatically. (lines 89, 93-94)

  1. Materials and Methods part indicates the determination of adrenaline, norepinephrine, dopamine levels in the subjects, but results are not presented. The validity of serotonin/creatinine score values references of in childhood (in different age groups) should be presented. Given the large age differences, it is necessary to provide data for different age periods. Moreover, serotonin levels were significantly higher in the first 12-13 years of life than in adolescents.

Answer 4. Again, we agree with your observation, Catecholamines are not part of this study. I have corrected the mistake. We included studies on cabbage and the corresponding statement. Additionally, I have added references regarding the validation of the serotonin/creatinine score.

  1. Metabolic syndrome in children is not clarified in the data: what is the patients BMI (allowed to
    be classified as metabolic syndrome).The role of metabolic syndrome in the study is unclear: are there some correlation of serotonin GIT level and clinical gastrointestinal and neuropsychiatric manifestations (data also should be presented).

Answer 5. Thank you for the amendment. The primary objective of this study is to implement a crucial preventive intervention for children. I have included additional information.(lines 294-297, 301-307

  1. Researchers stated in the study the exploration the modulation of the microbiota through
    probiotic therapy but also its modulation through dietary interventions. So no any data about
    microbiota and diet in the study.

Answer 6. Thank you very much for the observation. I completed with with clarifications (113-114)

  1. The polyprobiotic complex, diet used for intervention, are not clearly presented by researchers:
    data about strains of probiotics used with a proven psychobiotic effect (included in the
    composition), duration of therapy are not presented in the study.

Answer 7. Thank you for the thorough and highly intriguing observation. We have added more references about the effect of the probiotic on the gut-brain axis, and we have added the recommended period of use of the probiotic.

  1. The results of the study part presents data on the level of serotonin in the groups of
    gastrointestinal manifestations and neuropsychiatric manifestations, but the data of adrenaline-norepinephrine-dopamine.

Answer 8. Thank you for comment. In this paper, our aim is to track alterations in gastrointestinal and neuropsychiatric manifestations based on serotonin levels. Thank you for your feedback; I have rectified the error.

  1. The discussion of the results of the study is not connected with the practical part ; there are no
    clinical parallels of the data obtained on serotonin levels in the intestine and clinical
    manifestations with analogues in the studies, no data about elevated morning cortisol levels
    exhibited positive correlations with headaches and aggressive behavior in the practical part.
    It is recommended to modificate and finalize the article with subsequent publication.

Answer 9. Thank you for suggestion.We the authors, have revised the discussions to concentrate on the findings in comparable studies.

Reviewer 2 Report

Comments and Suggestions for Authors

The title:  I suggest changing the title to: Variety of serotonin levels in pediatric gastrointestinal disorders.

Abstract : Define the purpose of the work and conlusions more clearly.

Introduction: The content of the chapter should be shortened and focused on changes of the serotonin pathway in the gastrointestinal tract.

Material and methods:  No control group. Unclear exclusion criteria, because  the assessed symptoms occur in IBS. There are no criteria for severity of abdominal and mental symptoms. No details on nutritional intervention, i.e. probiotic treatment.

Results: Some tables and figures reguire more detailed description.

Discussion: Attention should be paid to the numerous publications on serotonin, its receptor and varied effects in the digestive tract.  Knowledge should be supplemented regarding the contribution of dietary factors and the microbiome on serotonin homeostasis.

Conclusions: Only the most important conclusions should be provided in accordance with the objectives of the work and the results obtained.

Author Response

Reviewer 2

Firstly, we, the authors of the present manuscript wish to thank you for thoughtful commentary you have provided to improve the quality of the paper. We are very grateful for the time and effort you have devoted to this task. We have extensively revised my manuscript according to the recommendations. All changes in the text and the new figures that we have redesigned are highlighted. Please, see the point-by-point answers to your comments below. All correction was highlighted in the manuscript.

  1. The title:  I suggest changing the title to: Variety of serotonin levels in pediatric gastrointestinal disorders.

Answer 1: Thank you for suggestion. We have changed the title.

  1. Abstract : Define the purpose of the work and conlusions more clearly.

Answer 2: Thank you very much for observation. I completed with detailed description. (lines 13-15, 20-22)

  1. Introduction: The content of the chapter should be shortened and focused on changes of the serotonin pathway in the gastrointestinal tract.

Answer 3: Thank you for amendmentI included additional information regarding changes in the serotonin pathway in the gastrointestinal tract.

  1. Material and methods:  No control group. Unclear exclusion criteria, because  the assessed symptoms occur in IBS. There are no criteria for severity of abdominal and mental symptoms. No details on nutritional intervention, i.e. probiotic treatment.

Answer 4: Thank you for observation. I corrected and completed the Material and methods section.

  1. Results: Some tables and figures reguire more detailed description.

Answer 5. Thank you for observation. We, the authors, completed each description.

  1. Discussion: Attention should be paid to the numerous publications on serotonin, its receptor and varied effects in the digestive tract.  Knowledge should be supplemented regarding the contribution of dietary factors and the microbiome on serotonin homeostasis.

Answer 6: Thank you for suggestion. We, the authors, have revised the discussions to concentrate on the findings in comparable studies.

  1. Conclusions: Only the most important conclusions should be provided in accordance with the objectives of the work and the results obtained.

Answer 7: Thank you very much for the comment. The conclusions were modified and condensed to align with the objectives.

Round 2

Reviewer 2 Report

Comments and Suggestions for Authors

Text completion is appropriate.